# Transparency Assessment on Level 2 Automated Vehicle HMIs

Yuan-Cheng Liu [1,*], Nikol Figalová [2] and Klaus Bengler [1]

1   Chair of Ergonomics, Technical University of Munich, Boltzmannstr. 15, 85748 Garching, Germany
2   Clinical and Health Psychology, University of Ulm, Albert-Einstein-Allee 41, 89069 Ulm, Germany
*   Correspondence: yuancheng.liu@tum.de; Tel.: +49-162-215-3303

**Abstract:** The responsibility and role of human drivers during automated driving might change dynamically. In such cases, human-machine interface (HMI) transparency becomes crucial to facilitate driving safety, as the states of the automated vehicle have to be communicated correctly and efficiently. However, there is no standardized transparency assessment method to evaluate the understanding of human drivers toward the HMI. In this study, we defined functional transparency (FT) and, based on this definition, proposed a transparency assessment method as a preliminary step toward the objective measurement for HMI understanding. The proposed method was verified in an online survey where HMIs of different vehicle manufacturers were adopted and their transparencies assessed. Even though no significant result was found among HMI designs, FT was found to be significantly higher for participants more experienced with SAE Level 2 automated vehicles, suggesting that more experienced users understand the HMIs better. Further identification tests revealed that more icons in BMW's and VW's HMI designs were correctly used to evaluate the state of longitudinal and lateral control. This study provides a novel method for assessing transparency and minimizing confusion during automated driving, which could greatly assist the HMI design process in the future.

**Keywords:** automated driving; human-machine interface; transparency; assessment method

## 1. Introduction

Automated vehicles are considered a revolutionary technology that could relieve human drivers from tedious and long-distance drives. In recent years, with the ability to execute both lateral and longitudinal controls, Level 2 driving automation [1] has been commercially available and become more and more prevalent. However, the system's safety could be compromised easily when the role transitions of human drivers and the capabilities and limitations of the automated systems are not well understood. The first fatal accident involving an automated vehicle gives us essential insight into this issue [2]. Despite being initially concluded as a "driver error", some argued that it is more likely a "designer error" and is possibly owing to the lack of clear boundaries to allocate the responsibilities during driving between human and automated systems [3].

The duty allocation between human and automated systems should be transparent. In the definition of SAE Level 2 automated vehicle (L2 AV), human drivers are required to supervise the Automated Driving System (ADS) and be ready to intervene and perform the remaining driving tasks not performed by the ADS when the ADS is engaged [1]. Hence, to guarantee driving safety when Level 2 ADS is engaged, human drivers should be well informed of the current ADS status. In other words, L2 AVs should continuously provide drivers with the necessary information so that a correct and effortless understanding of AVs' capabilities and safe take-over maneuvers can be achieved [4–6].

Before fully autonomous vehicles are generally adopted, human drivers must rely heavily on the Human-Machine Interface (HMI) to learn about the system state and take the corresponding action when a system boundary is reached. However, there has not been a standardized evaluation procedure or training session to guarantee that human drivers understand the HMI correctly. Furthermore, studies show that a significant number of ADS

drivers did not receive any information regarding the ADS, and the situation worsens for used-car owners [7]. Without correct information, proper training, or enough transparency regarding the ADS, the system capabilities and limitations would be unclear to drivers and could easily lead to more confusion and misuses [8].

Hence, the HMI design must be transparent to support unerring understanding of the automation system status. Naujoks et al. [9] proposed an AV HMI design guideline which aggregates HMI design recommendations from experts and uses heuristic verification as the evaluation method. However, the transparency of the HMI to human drivers is not addressed. It is also pointed out that a standardized test protocol for validation is not yet available [9]. Similarly, in the fields of robotics and vehicle automation, transparency for human-machine interaction has been emphasized [10–13], but a systematic and standardized evaluation method for transparency does not exist.

In this study, a transparency assessment method is proposed as a preliminary step toward a standardized validation protocol. The proposed method is aimed to facilitate a more efficient HMI design process and guarantee driving safety by evaluating the transparency of the HMI. Possible suggestions for a more transparent HMI design were also identified during the course to help increase the transparency of future HMI designs.

*Transparency*

Transparency of the HMI design has been studied across various ADS levels, and it is considered a basis for trust and acceptance [11,13,14]. Human drivers must develop a correct understanding of the ADS, which in turn supports driving safety. Maarten Schraagen et al. [15] evaluated the effects of providing transparency, post-hoc explanations, or both using videos of Level 2 ADS in different driving conditions. Results showed increased trust, satisfaction, and situational awareness when a moderate level of transparency is provided. Körber et al. [14] investigated whether explaining the take-over request in Level 3 ADS would result in different transparency, trust, or acceptance of the HMI. The study was carried out in a driving simulator. The subjective evaluation of system understanding did suggest that the HMI with explanations could increase transparency. However, further research is necessary to validate the actual improvement of system understanding.

Moving to Level 4 ADS, Pokam et al. [10] defined and divided the transparency into two levels: Robot-to-Human transparency and Robot-of-Human transparency. The former categorizes the information that a robotic system should convey to a human. Contrarily, the latter represents information about the awareness and understanding of the human that the system receives and presents to the human. The authors of the study were interested in the effect of HMI transparency on situational awareness, discomfort feelings, and participants' preferences. Five different HMI designs with varying levels of transparency were evaluated in the driving simulator. It was found that the transparent HMI provides a better understanding of the surrounding, and the participants would prefer higher transparency. In contrast to the study by Korber et al., who measured transparency by rating scales, Pokam et al. defined subjectively different transparency levels as the combinations of information provided on the HMI.

In another study on Level 4 ADS, the system transparency information was treated as a constant across different HMIs [11]. The authors of this study compared four types of interfaces and two types of transparency information (presentation of hazard and intended driving path) using the Wizard-of-Oz paradigm [16]. With three seven-point scale questions [17], the results showed increased system transparency in all three interfaces compared to the baseline one.

In the field of robotics, Chen et al. [18] proposed the situation awareness-based agent transparency (SAT) model, based on Endsley's situation awareness model [19], and categorized three levels of transparency based on the information type. In the first level, information about the current state and goals of the automation is provided; followed by the information regarding the reasoning behind the action in the second level; and finally, human users are supplied with predictive information in the third level. Yang et al. [20]

applied the definition of transparency in SAT to investigate the effect of automation transparency on human operators' trust. Two types of alarms with different transparency levels were designed. The traditional one (less transparent) uses a binary alarm, while the other one (more transparent) uses a likelihood alarm to provide additional information regarding the confidence and urgency level.

However, the definitions of transparency in the literature mentioned above are either subjective or merely used to categorize the information types, which do not disclose the driver's (or user's) understanding and efforts applied to the HMI. Additionally, a systematic and standardized evaluation method for estimating the HMI's transparency is not addressed. Without a standardized evaluation method, researchers could only identify a more transparent HMI design through comparison and were not able to differentiate in detail which components in the HMI design are critical to transparency. It would be even more challenging to estimate the effect of transparency on other measurements, such as trust and acceptance, when there is no standardized evaluation method. Moreover, a transparent HMI design should not solely be a combination of information topics since the information needed to understand the system differs across automation levels, traffic situations and driver characteristics [4,21–23]. In other words, giving the same HMI to different participants under different scenarios would result in different "functional transparency" for each individual. Providing more information (e.g., a higher transparency level in SAT model) is not necessarily what human drivers need. This was demonstrated by Carsten and Martens [4], who concluded that human drivers with more trust in the system tend to prefer HMI with less information. The differences in the information needed also make it more urgent to have a transparency assessment method, so the critical information leading to better understanding could be identified and applied to enable transparent HMI design.

Hence we argue that a standardized assessment method for functional transparency should exist so that the information needs under various conditions and driver characteristics can be identified efficiently and facilitate a transparent HMI design. Furthermore, the exact requirement for HMIs to facilitate correct understanding and minimum efforts could be identified by adopting the transparency assessment method. Here we outline the objectives of this study:

- A standardized and robust transparency assessment method would be proposed.
- Verification of the proposed method would be conducted using commercially available HMI designs.
- Information critical to HMI designs' functional transparency would be identified using the proposed method.

Based on the objectives, we defined the research questions and hypotheses as follows:

- Q1: How sensitive is the proposed transparency assessment method when evaluating different HMI designs and ADS experiences?
    - H1a: There is significant difference in functional transparency among different HMI designs.
    - H1b: There is a significant difference in functional transparency among participants with different ADS experiences.
- Q2: How does the proposed functional transparency relate to self-reported transparency?
    - H2 The higher the functional transparency, the higher the self-reported transparency.
- Q3: How is the information used by participants with different levels of functional transparency?
    - H3: Participants with different levels of functional transparency use different information sources when estimating system states.

To further corroborate the internal and external validity of the proposed assessment method, we evaluate whether different ADS experiences affect the understanding of HMIs

in Hypothesis 1b to obtain internal validity. Although the proposed method uses only objective data, self-reported data is also included in this study and compared to the proposed method in Hypothesis 2 to establish external validity.

## 2. Materials and Methods

In the literature on human-machine interaction, the transparency level of the HMI is often manipulated as the amount of information provided to humans [10,14,24]. However, as described in the previous section, the information must vary across participants and scenarios. The HMI transparency level in the literature does not reflect human drivers' actual understanding of the environment and might thus diminish safety during automated driving. Hence, we developed a transparency assessment method to evaluate human drivers' true understanding of the environment facilitated by the HMI. In this section, we first define the transparency we attempt to evaluate. Then the study design comparing different HMIs is introduced. Finally, the analysis of the collected data is presented.

### 2.1. Definition of Transparency

During the interaction between humans and automation, shared goals can only be achieved when there is "a harmonization of control strategies of both actors towards a common control strategy" [25]. This suggests that the ADS should be functional, transparent, and understandable to human drivers in the context of human drivers and automated vehicles. The term functional transparency (FT) is used to be distinguished from transparency in the literature. Note that the transparency in the literature solely represents the information provided by the HMI, while the FT is considered as the resulting understandability of the HMI after the interaction with humans. To evaluate and gain deeper insights, here we list the three basic requirements for the FT.

The first and fundamental one is that the HMI should enable a correct understanding of the ADS states (i.e., minimum mode confusion). As pointed out in the HMI guidelines of Naujoks et al. [9], the HMI should inform human drivers of the current ADS mode and the system state changes. For Level 2 and Level 3 ADS, the responsibility of human drivers during automated driving is constantly changing. The role of human drivers could be passengers during Level 3 drivings and suddenly become drivers when the system reaches its boundary. Hence, clear indications of the system states are indispensable for human drivers to avoid critical situations and to increase the FT.

The second requirement is that human drivers should be well informed of what automated functions could be used and how they should be adopted. Cao et al. [26] emphasizes the importance of "*user awareness*", which represents the user's understanding of "the available and possible automated driving modes, of the currently active mode and transitions among driving modes". ADS with higher FT should permit correct activation and transition among ADS modes.

The last requirement for FT is also stressed in the literature, where researchers argue that HMI should be efficient and easy to understand (i.e., minimizing the workload) and without confusion so that drivers can stay focused on the road and reduce the response time [4,26].

Combining all three requirements, we define FT as:

> how easy it is for users to understand and respond to ADS correctly

More than just a definition is required to develop a standardized transparency assessment method. Quantifiable measurements would be indispensable to make the assessment efficient during evaluation and analysis. When evaluating user experience, self-report measurement is a standard method to scale the constructs like usability, acceptance, and trust [27–29]. However, self-report measures only account for participants' preferences, which might not reflect how easy or successful the interaction with the system is [30].

A novel way to formulate the FT is proposed to approach this issue by including the concept of understandability. In computer science, it is defined as "the capability of the

software product to enable the user to understand whether the software is suitable, and how it can be used for particular tasks and conditions of use" [31]. By adopting the method of estimating code understandability [32], the formulation for FT is

$$T_{functional} = \begin{cases} 0, & \text{if "No" is answered} \\ AU(1 - \frac{TNPU}{TNPU_{max}}), & \text{otherwise} \end{cases} \tag{1}$$

where $AU$ represents the actual understandability, which is acquired through verification questions about the states of the HMI. Since $AU$ is calculated by the percentage of correct answers, it would be a number between 0 and 1. $TNPU$ stands for the time needed for perceived understandability, which is the time required by participants to state whether they understand the HMI or not. $TNPU_{max}$ is the maximum $TNPU$ measured within the targeted HMI cluster and is used to normalize the $TNPU$.

With the proposed formula, the FT of the HMI could be easily estimated from simple experimental setups.

### 2.2. Study Design

The FTs of a series of HMI images of Level 2 ADS from different brands and scenarios were evaluated to verify the proposed transparency assessment method. Besides the difference in HMI brands, different levels of experience in Level 2 ADS were also considered. The study was carried out on the online survey tool LimeSurvey.

#### 2.2.1. HMI Designs

In this study, HMI designs from BMW (Bayerische Motoren Werke AG, Germany) 3 Series, Tesla (Tesla, Inc, USA.) Model 3 and VW (Volkswagen AG, Germany) Passat were employed. The HMI images used were from the driver's perspective, where the instrument clusters were shown for the images of BMW and VW, while the touchscreen next to the driver was used for Tesla's images. The HMI designs on the market were chosen for their distinct design concept, where VW keeps the traditional layout (i.e., speedometer and tachometer) and puts system status on the bottom with relatively smaller icons (Figure 1). On the other hand, Tesla provides detailed information for ego and surrounding vehicles and takes traditional meters away (Figure 2). The HMI design for BMW combines the above two (Figure 3). Furthermore, using existing HMIs could make the results of this study a valuable foundation for future studies concerning various user experiences and mental models of different HMI designs.

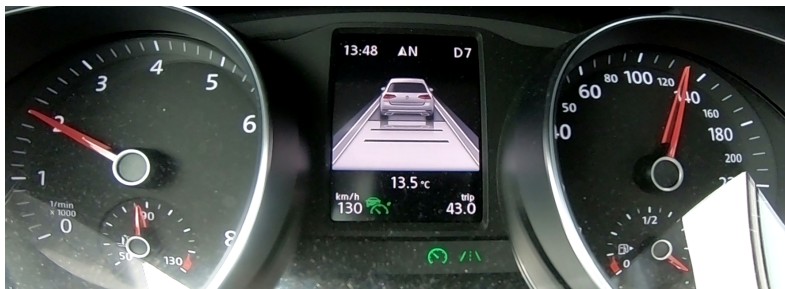

**Figure 1.** Example VW HMI design.

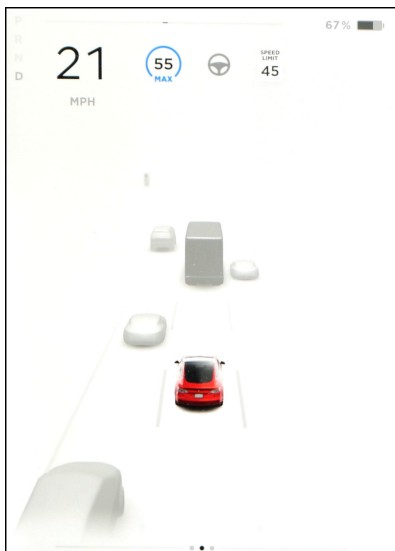

**Figure 2.** Example Tesla HMI design.

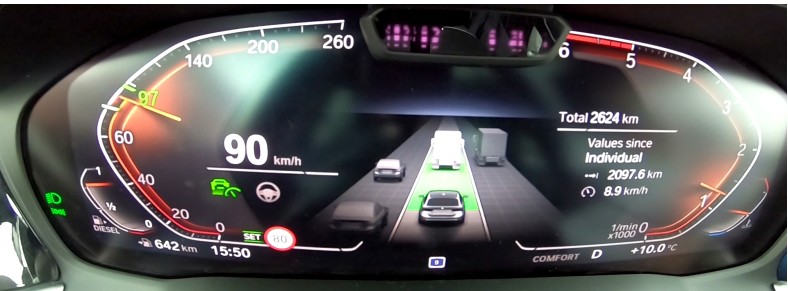

**Figure 3.** Example BMW HMI design.

Besides different brands, various HMI images under different scenarios were also considered. On Level 2 ADS, longitudinal and lateral control was supported by adaptive cruise control (ACC) and lane-keeping assistance (LKA). Whether these sub-systems are engaged or not, icons on the HMI designs would have different effects, resulting in various scenarios. Together with "whether the front vehicle is detected" and "if the Level 2 ADS could be activated", all possible HMI images for different scenarios are listed in Table 1. Note that similar or even the same icons might have different meanings across different HMI designs, so the availability of sub-systems is used to avoid confusion. With 11 different scenarios for each brand, a total of 33 HMI images are adopted in this study.

**Table 1.** HMI scenarios under different conditions.

| Scenarios | Is ACC Available? | Is LKA Available? * | Is Front/Side Vehicle Visible? | Is There Warning Signal? |
|---|---|---|---|---|
| Nothing activated | Yes | No | No | (None) |
| | Yes | No | Yes | (None) |
| | Yes | Yes | No | (None) |
| | Yes | Yes | Yes | (None) |
| Only ACC activated | (activated) | No | No | (None) |
| | (activated) | No | Yes | (None) |
| | (activated) | Yes | No | (None) |
| | (activated) | Yes | Yes | (None) |
| ACC and LKA activated (Level 2) | (activated) | (activated) | No | No |
| | (activated) | (activated) | Yes | No |
| | (activated) | (activated) | (None) | Yes |

* Note: No icon on BMW's HMI design indicates if the LKA is available. Hence, for the design, this column becomes " Is LKA on standby?".

### 2.2.2. Transparency Assessment Test

A transparency assessment test (TRASS) was employed after each HMI image was examined to estimate the FT. The critical questions to assess FT that are included in the TRASS should depend on the scenario the researchers would like to test. In this study, we mainly focused on the HMI of SAE Level 2 automated vehicles, in which the role of the driving automation system by definition is "Performs part of the DDT by executing both the lateral and the longitudinal vehicle motion control subtasks" [1]. Hence the states of these two sub-systems (longitudinal and lateral control systems) are critical to users and should be clearly transmitted. Together with experts' perspectives and literature [4,8,9,33], the following questions are critical for drivers to acquire a correct understanding of the Level 2 ADS:

1. Is the driving assistance system carrying out longitudinal control?
2. Is the driving assistance system carrying out lateral control?
3. Is the front vehicle detected by the driving assistance system?
4. Is the lane marking detected by the driving assistance system?
5. Can you activate the automated driving assistance (which performs both longitudinal and lateral controls automatically)

Each formal TRASS briefly introduced information regarding Level 2 ADS and its sub-systems (ACC and LKA). Participants were then instructed to answer questions regarding the ADS states based on the upcoming HMI image. After the image was shown, Participants were asked to choose either "Yes, I understand" or "No, I do not understand" based on whether they felt confident answering the HMI image's ADS states and were required to make the decision as fast as possible. Then, five questions regarding ADS states mentioned above were asked with three options: "Yes", "No" and "Uncertain".

The answers to these questions allow us to estimate the actual understanding of the participant regarding the ADS states, which is the AU in Equation (1). On the other hand, the time used to comprehend the HMI, which is the TNPU in Equation (1), is calculated by the time needed by participants to choose either "Yes, I understand" or "No, I do not understand", depending on whether they understood the HMI image and considered themselves capable of answering the questions mentioned above.

### 2.2.3. Self-Reported Transparency Test

A self-reported question regarding perceived transparency was asked on a 5-point Likert scale. The question referred to whether the information provided by the HMI was easy to understand ("With the information provided by the HMI image shown, do you agree that 'this HMI is easy to understand'"), which was later compared to the proposed transparency assessment method.

### 2.2.4. Information Used Test

The HMI image from each brand having its icons labeled with numbers was shown to the participants. Participants were then asked to identify the icons they used to answer the questions in TRASS: whether ADS is carrying out longitudinal or lateral control, whether front vehicle or lane marking is detected, and whether Level 2 ADS is activated. Both functional and irrelevant icons are labeled with numbers in each HMI image.

### 2.2.5. Procedure

In total, 33 individuals participated in the $2 \times 3$ mixed design. The within-subject factor was three different brands of HMI designs: (1) BMW, (2) Tesla, and (3) VW. Experience in ADS was the between-subject factor in the two groups. Regarding the question "How often have you used the following driving assistance systems in the past 12 months: cruise control, adaptive cruise control, lane-keeping assistance, and automated driving assistance?" those who answered "used sometimes" or "used regularly" are categorized as "experienced" ($n = 16$), and the rest ("rarely used", "known but never used" and "unknown")

as "novice" ($n = 17$). Regarding the experiences of certain vehicle brands, among those characterized as "experienced" ($n = 16$), nine of them are experienced in VW's ADS, four in BMW's ADS, and four in Tesla's ADS. Participants were balanced for gender (18 males and 15 females) and age ($M = 29.48, SD = 6.00$), and had been driving for 3 years or more ($M = 10.61, SD = 6.47$).

After collecting demographic data, calibration tests were conducted to mitigate possible internet delays and individual reaction time differences. The reaction time was later used for calibration in the following analysis. Participants were informed to click either "Yes, I understand" or "No, I do not understand" (randomly assigned) on a dummy figure after pressing the next button. Then, the dummy figure was shown with both options below, which is the same procedure as in TRASS. Practice tests were also presented before the formal TRASS to familiarise the experimental process. The same design and layout of formal TRASS were introduced in the practice test, but still, the section for the HMI image was replaced with a blank figure to avoid any bias.

During the formal TRASS, ten tests were executed for each participant and took around 30 to 40 mins to complete. From 33 HMI images with different HMI brands and scenarios, ten images were randomly chosen for each participant. The process described earlier for TRASS was followed, and a self-reported transparency test was conducted at the end of each TRASS. Finally, the information used test was performed for each brand at the end of the survey.

### 2.3. Analysis

2.3.1. Transparency Assessment Test

Using the proposed Equation (1), the functional transparency (FT) for each TRASS was calculated with the reaction time (TNPU) and the actual understanding (AU) collected during the test. To determine whether the HMI brands and Level 2 ADS experiences have any effect on transparency, we performed the linear mixed effect analysis with lme4 [34] in the R-4.2.1 programming environment [35]. HMI brand and Level 2 ADS experience were treated as fixed effects in the model, with their interaction term also considered. For random effects, intercepts for participants and scenarios were set, but no by-participant or by-scenario random slopes for either fixed effect could be added as the model could not converge. The representation for the final maximal structure [36] model is:

$$FT \sim HMI\ brands * Level\ 2\ ADS\ experience + (1\,|\,participant) + (1\,|\,scenario) \qquad (2)$$

No apparent deviations from homoscedasticity or normality were found with a visual inspection of residual plots. Likelihood ratio tests were applied to models with and without the targeted effect. Further pairwise comparisons were conducted using emmeans [37] also in the R programming environment, where Kenward–Roger degrees-of-freedom approximation and Tukey adjusted *p*-value were applied.

2.3.2. Self-Reported Transparency Test

The responses were collected and compared to the proposed FT. Their relationship would be determined using Spearman's correlation.

2.3.3. Information Used Test

Results of icons used to answer each question in the TRASS were collected. Participants with different levels of FT (higher and lower) and their responses to different HMI designs (BMW, Tesla, VW) were analyzed using mixed ANOVA in JASP 0.14.1.0. The post hoc Holm-Bonferroni test was also carried out for comparisons among groups.

### 3. Results

*3.1. Transparency Assessment Test*

Table 2 shows the means and standard deviation of FT and self-reported transparency given the HMI designs and ADS experience levels. The data reveal that Self-reported transparency is higher than FT across all conditions. Furthermore, with Equation (2), the corresponding coefficients could be determined. Table 3 shows estimates and standard error of fixed effects, and 95% confidence intervals (abbreviated as 95% Conf. Int.) for the estimates. We are 95% confident that participants experienced in ADS have 0.027 to 0.22 higher FT than novice ones. Finally, Using the linear mixed-effect model with likelihood ratio test, the results indicate that three different HMI designs have no significant effect on FT, $\chi^2(2) = 1.18, p = 0.56$. However, the effect of ADS experience levels on FT was found significant, $\chi^2(1) = 9.64, p = 0.02$, where experienced participants had $0.12 \pm 0.048$ higher FT than novice participants. No significant interaction effect between HMI designs and ADS experience levels was found, $\chi^2(2) = 2.33, p = 0.33$.

**Table 2.** Mean and standard deviation (SD) of measured FT and self-reported transparency with different HMI designs and ADS experience levels.

| HMI Design | ADS Experience Level | Functional Transparency Mean (*SD*) | Self-Reported Transparency Mean (*SD*) | N |
|---|---|---|---|---|
| BMW | experienced | 0.45 (0.26) | 0.67 (0.23) | 52 |
|  | novice | 0.32 (0.24) | 0.71 (0.21) | 54 |
| Tesla | experienced | 0.41 (0.24) | 0.56 (0.24) | 51 |
|  | novice | 0.34 (0.24) | 0.59 (0.23) | 60 |
| VW | experienced | 0.42 (0.21) | 0.62 (0.20) | 59 |
|  | novice | 0.29 (0.21) | 0.58 (0.19) | 54 |

Note: The experience level here is regarding the general ADS experience, and not specifically for a certain HMI design.

**Table 3.** Estimates of fixed effects for the mixed effect model.

| | Estimate | Std. Error | df | t Value | Sig. | 95% Conf. Int. | |
|---|---|---|---|---|---|---|---|
| | | | | | | Lower Bound | Upper Bound |
| Intercept | 0.45 | 0.040 | 65.11 | 11.24 | <0.0005 | 0.37 | 0.52 |
| ADS experience: exp. | 0 | 0 | . | . | . | . | . |
| ADS experience: novice | −0.12 | 0.048 | 122.01 | −2.54 | 0.012 | −0.22 | −0.027 |
| HMI design: BMW | 0 | 0 | . | . | . | . | . |
| HMI design: Tesla | −0.035 | 0.043 | 309.08 | −0.80 | 0.43 | −0.12 | 0.051 |
| HMI design: VW | −0.018 | 0.041 | 302.56 | −0.44 | 0.66 | −0.10 | 0.063 |

Note: Fixed effects of interactions are not listed.

Further comparisons are shown in Figures 4 and 5. In Figure 4, FT from the same HMI design were grouped, where participants with different experience levels were compared. The result indicates that, more experienced participants showed significantly higher FT in BMW HMI, $t(127) = 2.50, p = 0.014$, and VW HMI, $t(118) = 3.06, p = 0.003$. However, the effect of different ADS experiences levels on Tesla HMI design was insignificant, $t(121) = 1.26, p = 0.21$. In Figure 5, given the same ADS experience level, no significant effect was found across comparisons between different HMI designs. Results of all the pairwise comparisons are listed in Table 4.

**Table 4.** Results of all pairwise comparisons.

| Given Variable | Comparison | Estimate | SE | DOF | t-Value | *p*-Value |
|---|---|---|---|---|---|---|
| HMI design: BMW | exp - novice | 0.12 | 0.049 | 127 | 2.50 | 0.014 ** |
| HMI design: Tesla | exp - novice | 0.06 | 0.048 | 121 | 1.26 | 0.21 |
| HMI design: VW | exp - novice | 0.14 | 0.047 | 118 | 3.06 | 0.003 *** |
| ADS experience level: exp | BMW - Tesla | 0.035 | 0.044 | 313 | 0.79 | 0.71 |
| | BMW - VW | 0.018 | 0.042 | 307 | 0.43 | 0.90 |
| | Tesla - VW | −0.017 | 0.042 | 305 | −0.39 | 0.92 |
| ADS experience level: novice | BMW - Tesla | −0.027 | 0.042 | 318 | −0.66 | 0.79 |
| | BMW - VW | 0.040 | 0.043 | 320 | 0.94 | 0.62 |
| | Tesla - VW | 0.068 | 0.041 | 305 | 1.65 | 0.23 |

*** $p < 0.01$, ** $p < 0.05$.

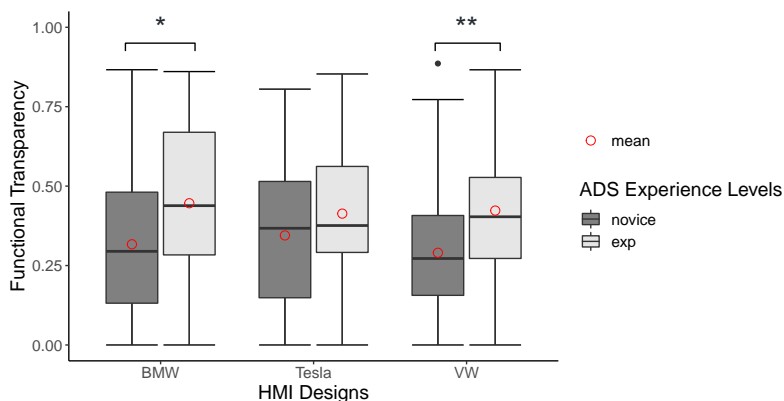

**Figure 4.** Comparing levels of ADS experience given HMI designs (exp stands for experienced in ADS experience levels). ** $p < 0.05$, * $p < 0.1$.

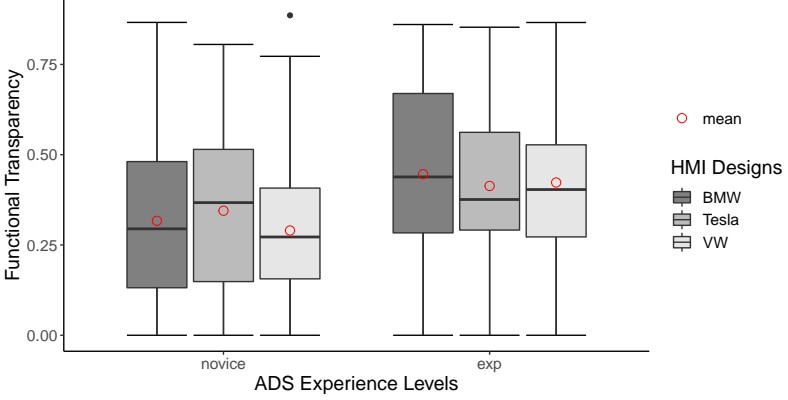

**Figure 5.** Comparing HMI designs given ADS experience levels (exp stands for experienced in ADS experience levels).

### 3.2. Self-Reported Transparency Test

The correlation between the proposed FT and self-reported transparency was calculated using Spearman's correlation ($r_s$). The self-reported transparency was normalized between 0 and 1, in the same range as the proposed FT. The result suggests that there was a weak monotonic relationship, $r_s = 0.25$, $p < 0.0001$, between the objectively measured FT and the self-reported transparency.

### 3.3. Information Used Test

To understand the differences in the information used, we divided the participants into high FT ($n = 16$) and low FT ($n = 17$) groups based on their average FT throughout the total of 10 TRASSes, where the median was chosen as the threshold ($Mdn = 0.39$).

The HMI designs are shown in Figures 6–8, having their icons labeled with a number. The corresponding icons chosen by participants to answer TRASS questions are shown as bar charts in Figures 9–11, with each subplot representing one of the TRASS questions: questions regarding longitudinal control (Long), lateral control (Lat), front vehicle (FV), lane marking (LM), and Level 2 ADS availability (Ava).

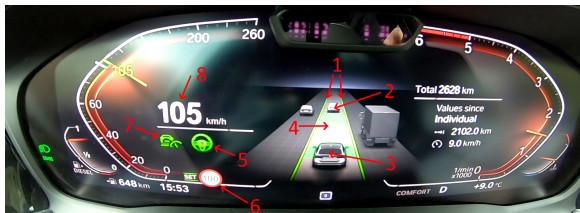

**Figure 6.** BMW HMI design and corresponding icon number.

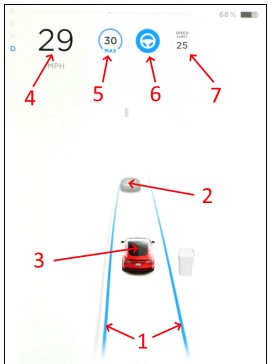

**Figure 7.** Tesla HMI design and corresponding icon number.

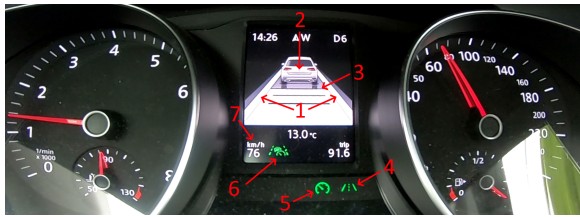

**Figure 8.** VW HMI design and corresponding icon number.

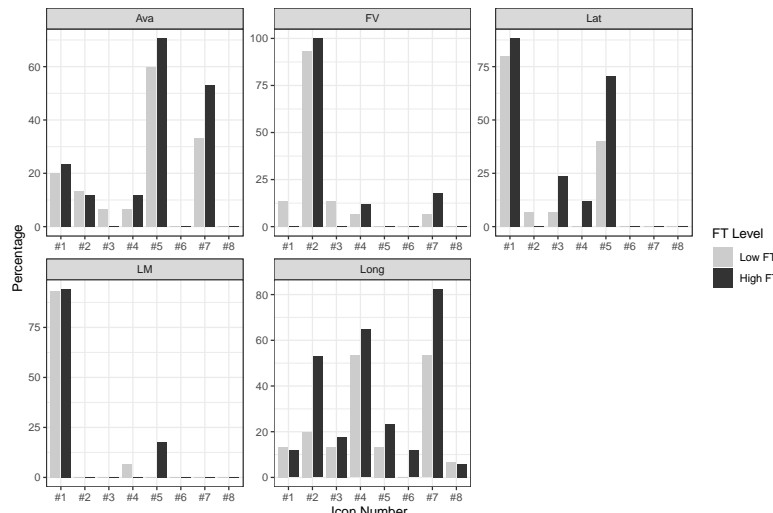

**Figure 9.** Percentages of BMW icons used to answer TRASS questions from participants with high and low FT.

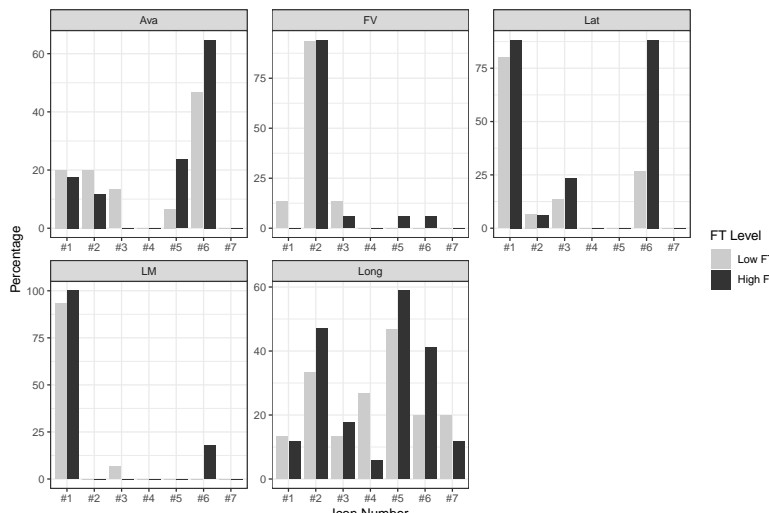

**Figure 10.** Percentages of Tesla HMI icons used to answer TRASS questions from participants with high and low FT.

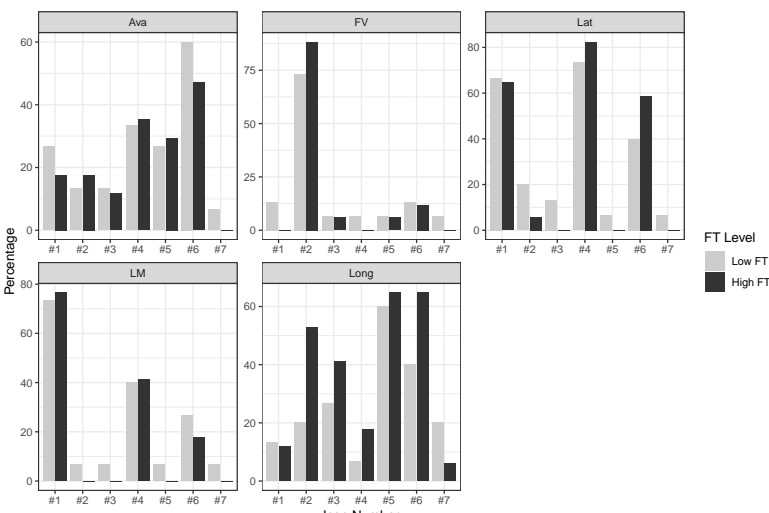

**Figure 11.** Percentages of VW HMI icons used to answer TRASS questions from participants with high and low FT.

As shown in Table 5, no valid icon could be used in BMW and VW's HMI designs to determine whether automated longitudinal and lateral controls could be activated. For Tesla, icon #6 is used to indicate the availability of level 2 ADS.

To determine whether front vehicle is detected, icon #2 is used in all HMI designs (Figures 6–8), and there is an extra icon for BMW (icon #7 in Figure 6). There was no significant difference on valid icon used for FT levels, $F(1,31) = 1.48$, $p = 0.23$, and HMI designs, $F(2,62) = 2.84$, $p = 0.067$, and there was also no interaction between these factors, $F(2,62) = 0.50$, $p = 0.61$. There were generally low false icon selection rates in answering questions regarding the detection of front vehicles for all three designs. Still, there was a relatively higher portion of participants choosing icon #6 in VW's design (Figure 8), where the small vehicle icon is shown regardless of whether the front vehicle is detected or not.

To answer the question concerning whether lateral control is activated, icon #1 and #5 are used on BMW's design (Figure 6), and icon #1 and #6 for Tesla and VW's (Figures 7 and 8). No significant effect was found on different FT levels, $F(1,31) = 1.00$, $p = 0.33$, but the effect of HMI design was found significant, $F(2,62) = 4.31$, $p = 0.018$, where more participants chose at least one valid icon on BMW's design comparing to VW's, $t(32) = 2.54$, $p = 0.041$, and same applies to Tesla's comparing to VW's, $t(32) = 2.54$, $p = 0.041$, but no difference

between BMW's and Tesla's design, $t(32) = 0.00, p = 1.00$. In Figure 11, icon #4 on VW (Figure 8) was chosen falsely by participants, and it indicates the activation of "lane assist", which would only intervene when the vehicle is about to cross the lateral boundary (lane marking).

In all HMI designs, lane marking detection was indicated by icon #1 (Figures 6–8), and additional icons were used on VW's design (#4 and #6 in Figure 8). The effect of FT level was not significant, $F(1, 31) = 0.26, p = 0.61$, and was the same for HMI designs, $F(2, 62) = 0.21, p = 0.81$, and for the interaction as well, $F(2, 62) = 0.21, p = 0.81$.

The question regarding if longitudinal control is activated could be answered by icon #2, #4 and #7 on BMW's, #5 on Tesla's, and #2, #3, #5 and #6 on VW's design, where no significant effect was found on different FT level $F(1, 31) = 0.73, p = 0.40$, but the effect of HMI design was significant, $F(2, 62) = 8.66, p < 0.001$, as more participants chose at least one valid icon on the HMI design of BMW than Tesla, $t(32) = 3.61, p = 0.002$, and same for VW's design comparing to Tesla's, $t(32) = 3.61, p = 0.002$, but not between BMW's design and VW's, $t(32) = 0.00, p = 1.00$. No interaction effect was found between the two factors, $F(2, 62) = 0.034, p = 0.97$. Although no effect was found for FT levels on the number of participants choosing at least one valid icon, higher percentages of valid icon selections were observed across all HMI designs, where for each valid icon, more participants with high FT selected it than participants with low FT.

**Table 5.** Valid and false icons on HMI designs concerning TRASS question categories.

| Question | HMI Designs | Valid Icons | False Icons |
|---|---|---|---|
| Ava | BMW | Unknown | Unknown |
| | Tesla | #6 | #1, #2, #3, #4, #5, #7 |
| | VW | Unknown | Unknown |
| FV | BMW | #2, #7 | #1, #3, #4, #5, #6, #8 |
| | Tesla | #2 | #1, #3, #4, #5, #6, #7 |
| | VW | #2 | #1, #3, #4, #5, #6, #7 |
| Lat | BMW | #1, #5 | #2, #3, #4, #6, #7, #8 |
| | Tesla | #1, #6 | #2, #3, #4, #5, #7 |
| | VW | #1, #6 | #2, #3, #4, #5, #7 |
| LM | BMW | #1 | #2, #3, #4, #5, #6, #7, #8 |
| | Tesla | #1 | #2, #3, #4, #5, #6, #7 |
| | VW | #1, #4, #6 | #2, #3, #5, #7 |
| Long | BMW | #2, #4, #7 | #1, #3, #5, #6, #8 |
| | Tesla | #5 | #1, #2, #3, #4, #6, #7 |
| | VW | #2, #3, #5, #6 | #1, #4, #7 |

Note: For BMW and VW, level 2 ADS could be engaged without guaranteed longitudinal and lateral control. Hence, no valid icon could be used to answer the question "Ava".

## 4. Discussion

### 4.1. Summary

HMI is the bridge that allows humans to understand the intentions and capabilities of ADS. At the same time, transparency of the HMI is critical and fundamental for the ADS to be understood with minimum effort [4]. In this study, a preliminary transparency assessment method was proposed, integrating the measurement of understandability toward mode awareness, and using time as the workload indicator. With the proposed method, the functional transparency of the static level 2 ADS system could be evaluated with minimum effort.

The results using the proposed transparency assessment method did not confirm Hypothesis 1a, but support Hypothesis 1b. During the verification test of the transparency assessment method, no significant difference was found among the three different HMI designs. This could be explained by the generally low FTs measured, suggesting that participants had difficulties understanding these three HMI designs. When the proposed

method is properly utilized and incorporated into the HMI design process, we could then more efficiently develop a more understandable HMI design and overcome this problem. For instance, critical elements that help increase HMI understandability could be efficiently identified by observing what information users with high FT use to understand it. And the opposite could also be done to identify potentially misleading information. On the other hand, the effect of ADS experience levels on FT was found significant and was significantly more prominent in the HMI designs of BMW and VW. The result supports Hypothesis 1b and establishes internal validity. This significant gap in understanding the HMI designs between experienced and novice users also suggests that human drivers require some levels of training to understand the HMI [33,38,39]. Apart from general ADS experience, it might also be interesting to look into how a specific brand's experience interacts with the HMI designs of some other brands on the FT. In the feedback section of the survey, some participants mentioned that they made specific suggestions based on their experiences in L2 AV. Since the design, icons, and logic behind the HMI designs are different, further studies investigating such interaction could be valuable in the future.

The subjective transparency estimated with one item Likert scale showed a weak positive correlation with FT. This result supports Hypothesis 2 and establishes external validity. The self-reported measurement was considered as the perceived transparency of participants in contrast to the proposed FT, which is the resulting understanding after the interaction. From the result, it appears that what participants thought they understood was detached from what they did. However, a certain level of monotonicity still exists between the two measurements, which gives an insight for future studies on the relationship between perceived transparency (what one thinks one knows) and functional, or true, transparency (what one actually knows).

What and how information should be conveyed through HMI has been an essential topic in automated vehicle research [4,9,21]. In this study, we evaluated the information used to understand level 2 ADS statuses (i.e., longitudinal control status, lateral control status, front vehicle detection status, lane marking detection status, and Level 2 ADS availability status). Overall, participants with higher FT levels did not choose icons more correctly. Still, in evaluating specific system statuses, they selected more valid icons, which suggests that participants with higher FT relied on multiple information sources when estimating these system statuses. And this result supports Hypothesis 3. On the other hand, HMI designs had no significant impact on icon identification across all system statuses, but it was found significant when evaluating longitudinal and lateral status. By looking closer, e.g., when assessing the longitudinal status, those with a higher percentage of at least one valid icon selection are HMI designs with a higher number of valid icons (BMW and VW's designs). This information redundancy in status indication might seem to be helpful, but it could also lead to confusion and wrong icon selection [40]. Again using the longitudinal status estimation as an example, the redundant icon for lane detection and "lane assist" (only intervene when the boundary is reached) in VW's HMI design (#4 in Figure 8) was mistaken as the indication for the lateral control. Hence, to achieve transparent HMI, it is more critical to provide information with quality (correct information given the level of automation and the scenario) instead of quantity (merely stacking the information). And the proposed method would be a suitable design tool to help identify critical information for transparent HMIs.

### 4.2. Limitations and Future Works

In this study, the differences in understandability of the adopted HMI designs are limited, which might also be the culprit for failing to confirm Hypothesis 1a. However, we also identify some elements of the HMI designs that have an impact on FT. With these elements, we could create and adopt HMI designs with more significant differences in understandability in future research.

The information needed for the HMI differs across levels of automation and scenarios [41]. This study focused on the level 2 HMI designs, where human users are still responsible for

longitudinal and lateral controls and driving environments. However, in a higher level of automation, human users are no longer in the loop under specific conditions and are allowed to conduct non-driving-related tasks. These differences in responsibility would also significantly impact information needed for FT, thus affecting the questions asked during the transparency assessment test. It is crucial first to ascertain the critical questions for human users to be capable of safely operating the automation under various levels. Furthermore, to apply the transparency assessment method more efficiently, a non-intrusive way of estimating mode awareness [42] could be applied and used to replace the AU.

To further validate the proposed transparency assessment method, simulator or test-track studies are required. In this study, the transparency assessment test was based on HMI images under different scenarios. However, the interaction between participants and the automation was limited. For instance, with only images, participants wouldn't be able to experience the transition between different modes (i.e., activation, deactivation, take-over, etc.) and the corresponding reactions from the vehicle. Plus, as automation users are prone to learn automation by trial and error [7], understanding how transparency changes throughout the interaction would also be beneficial for future HMI designs.

One finding in this study is that participants with different levels of level 2 ADS experience did differ in their understanding of the automated system. But since the HMI designs and the logic behind the system's activation or deactivation diverge and might contradict one another, a more precise distinction on ADS experience would be essential (e.g., familiarity across different automation brands). With this variable, additional system information or tailored training procedure could provide better transparency for the HMI.

This study was a preliminary attempt to assess transparency, and the relationships between the proposed metric and other psychometrics (e.g., trust, acceptance, mental workload) remain unknown. Shedding light on these correlations would help make the proposed transparency assessment method more robust and assist in comparing it with the results in the literature.

### 4.3. Conclusions

In this study, we proposed and verified a standardized transparency assessment method, which can be used to estimate the understandability of the HMI design. We further established this method's internal and external validity by confirming that the effect of ADS experiences was significant on functional transparency and that a positive correlation between self-reported understandability and functional transparency existed. The proposed method can also help identify critical elements in HMI designs that can significantly impact the understandability of the HMI.

**Author Contributions:** Conceptualization, Y.-C.L.; methodology, Y.-C.L.; software, Y.-C.L.; validation, Y.-C.L.; formal analysis, Y.-C.L.; investigation, Y.-C.L.; resources, Y.-C.L. and K.B.; data curation, Y.-C.L.; writing—original draft preparation, Y.-C.L.; writing—review and editing, Y.-C.L. and N.F.; visualization, Y.-C.L.; supervision, K.B.; project administration, K.B.; funding acquisition, K.B. All authors have read and agreed to the published version of the manuscript.

**Funding:** This project has received funding from the European Union's Horizon 2020 research and innovation program under the Marie Skłodowska-Curie grant agreement No. 860410.

**Institutional Review Board Statement:** The study was conducted in accordance with the Declaration of Helsinki and approved by the Ethics Committee of the Technical University of Munich (377/21 S-KH) for studies involving humans.

**Informed Consent Statement:** Informed consent was obtained from all subjects involved in the study.

**Data Availability Statement:** The survey and collected data are accessible at: https://github.com/mobydickhm1851/transparencyAssessmentMethod_openAccessData.git (accessed on 29 September 2022).

**Conflicts of Interest:** The authors declare no conflict of interest.

## Abbreviations

The following abbreviations are used in this manuscript:

| | |
|---|---|
| ACC | adaptive cruise control |
| ADS | automated driving system |
| AU | actual understandability |
| BMW | Bayerische motoren werke AG |
| FT | Functional Transparency |
| HMI | human-machine interface |
| L2 AV | SAE level 2 automated vehicle |
| LKA | lane-keeping assistance |
| SAT | situation awareness-based agent transparency |
| TRASS | transparency assessment test |
| TNPU | time needed for perceived understandability |
| VW | Volkswagen AG |

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
