# Peer review of "Transparency Assessment on Level 2 Automated Vehicle HMIs"

_information, doi:10.3390/info13100489_

Round 1
Reviewer 1 Report
Summary
The authors propose a method to measure transparency of HMI’s for Automated Driving Systems (ADS) that does not crucially rely on subjective ratings (using a rating scale), but still includes the understanding by users (instead of only considering the objective properties of HMIs, by which measure ADS would be more transparent, the more information they contain.
[Note: I realise that the terminology in this summary is confusing, since it does not clearly distinguish between transparency of a system (ADS) and transparency of an HMI. The manuscript suffers from the same confusion.]
The manuscript introduces the need for a structured method to measure transparency, describes the method and the procedure, and presents results. The results are discussed and limitations and future work are presented.
Comments
I compliment the authors with the laudable effort to give hands and feet to the notion of transparency, which receives much attention in the context of ADS. The paper reads well and is adequately structured, and the topic is of interest to the readership of the journal, so I am sympathetic to the idea that it will be published. However, I have a number of questions and comments, to be detailed below, that I think should be addressed, and that require major revisions of the manuscript before it can be published.
Major comments
If a new method is introduced, several requirements may be formulated: internal and external validity, sensitivity, applicability as a tool for design (that is, being instrumental in choosing between different designs). In the current context, one might say that internal validity concerns questions such as whether people who have experience with ADS find it easier to understand ADS HMIs than people who have no such experience (Q1, H1b). External validity concerns questions such as whether the outcomes of applying the method align with those of proven methods. One might say that the question about the correlation between the results of the proposed method and the results of the subjective ratings (Q2, H2) concern the external validity. I think it would help if the issues of internal and external validity are explicitly discussed in the introduction.
Furthermore, one might say that sensitivity is related to the issue of internal validity, and here the question is how to establish that the proposed method is sensitive enough to measure the phenomenon adequately. The authors introduce question Q1 (Line 122) concerning sensitivity and propose to answer this question by evaluating two hypotheses, one (H1a) about whether the method is able to measure differences in transparency between different designs, and the other (H1b) about whether different types of users, differing in experience with ADS, demonstrate differences in their understanding of the HMI. Unfortunately, H1a is not supported by the data. Now, are we going to conclude that the method is not sensitive? Or do we conclude that, other things (such as level of expertise) being equal, there are no differences in transparency between the HMIs? There is no way out of this, because potential differences in the understandability of the HMIs was not established a priori. In my opinion, this is a major weakness in the design of the study.
Further questions concern the formula to calculate Functional Transparency, the questions that people have to answer to show whether they understand the HMI, and the design of the study.
Concerning the formula: the authors propose that Functional Transparency (FP) is composed from the score on questions that address the understanding of the HMI and the time it takes to arrive at this understanding. My first question is whether alternative formulas were considered, and if so, which ones. If the aim of the paper is to propose a new method to measure transparency (which is in essence a design challenge), this is relevant information: which alternative designs were considered and what were the arguments for choosing this one. Secondly, one might argue that transparency is strongly related to the issue of usability. In turn, usability is made up of effectiveness (is the information communicated successfully?), efficiency (how much time does it take to communicate the information?) and satisfaction (which is irrelevant to the message I want to convey). Now, it is commonly accepted that effectiveness is of primary importance for novice users, while, for experienced users, for whom effectiveness may be taken for granted, efficiency is of primary importance. Therefore, it is not obvious, that our insight into transparency is furthered by a formula that combines effectiveness and efficiency alike for novices and experienced users. Maybe understandability (AU) and Time (TNPU) should be analysed and discussed separately? Moreover, it raises the question for whom we should design, for novices (as the authors state (Line 391), “the intuitiveness of the HMI designs is necessary to overcome this problem [of understanding … HMI designs]”), or for more experienced users. Maybe, different types of users need different designs? If so, will the proposed method help to arrive at this conclusion?
The method involves generating questions to probe people’s understanding of the HMI (Lines 227-232). Here, I had the question how these questions were generated. Again, this is a design problem. Are these all and only the relevant questions? How were they chosen? And to which level of detail does one need to go? For instance, about longitudinal control (which is basically a matter of ACC), does the user need to know whether it is full range or restricted range (>30 km/h) ACC? This is certainly a matter of transparency, but is this level of detail needed? More generally, what are requirements for transparency and how do we generate questions to probe whether a particular design is transparent or not?
Another question I had concerns the design of the study. Participants received 10 randomly chosen images out of 33 possible images showing different states of at most three HMIs, meaning that they (assuming a random distribution) saw three or four instances of the same HMI (showing different states of the HMI). One might expect that, after they saw a particular HMI once and learned what questions they had to answer, they knew what information to look for and where to look for it. Moreover, once they learned what questions they had to answer, they also knew what information to look for in the other HMIs. The authors might contend that this is not a problem, since the questions were precisely about the information that the HMIs aimed to communicate anyway, but one way or another it feels not fully correct to first inform people what information they have to look for and then, if they manage to find this information, to say that the method adequately measures transparency. At a minimum, I would like to see an approach where carry-over effects are removed from the data.
Concerning the composition of the sample, roughly half of the participants were experienced people. Now I would expect that there were some who had experience with BMW, some with VW, some with Tesla and some with yet other brands, and I would also expect that there were strong effects of prior knowledge. In the first place, the information about how many BMW, VW, Tesla and other experienced drivers there were, is not included. In the second place, it might be worthwhile to provide information about the effects of prior knowledge.
Minor comments
L232: “Can you activate …”: it was unclear to me what the precise meaning of this expression is. Does it mean: “given the HMI, would you know how to activate …”? Or does it mean: “Do you know whether this feature is off and can be turned on?” Please clarify.
L234: “Participants were then instructed to anticipate an HMI image”: what does ‘anticipate’ mean? What were they supposed to do: form some mental image without actually having seen an image? If so, what was the purpose of this instruction? And how does it work for novices?
L287: “impractical icons: What are ‘impractical’ icons?
Language/grammar/terminology/presentation
General point: In the end, we want people to understand the ADS; that is, the ADS should be transparent, and the HMI is a window to the system that should communicate the information about the system adequately and clearly. But do we use “transparency” also for the HMI? I find formulations talking about transparency of HMIs conceptually confusing, so please be careful about the terminology, and maybe explain why you use particular terminology. See e.g. lines 381 and 396: Line 381: is transparency critical for the HMI or for the system to be understood? Line 396: might human drivers need some training to understand the automation or the HMI?
L72: It is found > it was found
L102: I found the expression “information needs” confusing. I first read ‘needs’ as a verb, only afterwards realising that it was meant as a noun. I would prefer “information needed” here, also because “information needs to understand to system” sounds awkward. (And don’t forget to change ‘differ’ to ‘differs’ further down the sentence.)
L128: How the proposed … relates > How does the proposed … relate
L129: The higher … is, the higher > The higher …., the higher the (remove ‘is’ and insert ‘the’)
L131: How the information is used > How is the information used
L139: could not > does not
L187: Insert parentheses around Scalabrino et al. [31]
L220: are > is
L222: In the note to Table 1, the question mark is top down. Also, it might help to add a column in the table and number the scenarios from 1 to 11.
L308: data reveals > data reveal [data is plural!]
L351: regardless of front > regardless of whether
L353: to answer question > to answer the question
L359: in Fig 11 that icon #4 > remove ‘that’?
L367: question > the question
L467: the relationships > and the relationships
Author Response
Please see the attached PDF file.

Reviewer 2 Report
Dear authors
Your paper titled (Transparency Assessment on Level 2 Automated Vehicle HMIs) reveal a good work and highly relevant findings to the field of automobile automation. My overall evaluation is good and I believe that a further revisions would make the paper ready for publication. My comments are as follows:
-Title: for me and researchers in our field "level 2" is very clear. However, many readers may not aware of level 2. Would it be possible to clearly name it (partial driving automation) or for example (Transparency Assessment on Partially Automated Vehicle HMIs). Its up to you anyway.
- Abstract: (FT was significantly higher) this is unclear for me or the reader. Is it higher in a good way or a bad way. Particularly, what is higher, and higher than what.
- Small sample size: for a questionnaire-based method, 33 participants are not enough.
- Are all the participants from Germany? if yes, I think there could be a bias toward BMW and VW compared with Tesla. For example, if you conduct the study in the US, results would be bias toward Tesla.
- Did you inform the participants about the car brand to which the presented HMI belongs? This would make a different.
- I believe a conclusion section is essential. For example, you can add "4.3. Conclusions" in which you can present the main message/findings of the paper. Just one short paragraph is enough.
Author Response
Please see the attached PDF file.

Reviewer 3 Report
The authors tackle a fundamental topic in the field. Namely, how to assess the transparency of HMIs. In this respect, the authors make a distinction between “transparency” and “functional transparency” (FT), defining the latter as “How easy it is for the users to correctly understand and respond to ADS”. The assessment method revealed no differences between the tested HMI designs. However, functional transparency appeared higher for participants that had previous experience with Level 2 functionalities. I have read the article with interest. Nonetheless, I have some true concerns about the novelty of the approach and the significance of the contribution. Furthermore, other aspects of this manuscript should be carefully revised:
1) More literature should be included in the Introduction. In this respect, although the authors investigate transparency of Level 2 systems, section 1.1 only discusses studies investigating Level 3 and Level 4 HMIs. In general, it should be clarified how the transparency of Level 2 HMIs is currently assessed (e.g. Kraus et al., 2020; Oliveira et al., 2020).
2) A true definition of “Transparency” is lacking. The authors state: “Note that the transparency in the literature solely represents the information provided by the HMI, while the FT is defined as the resulting understandability of the HMI after interacting with humans” (151-154). This statement is vague, is not a definition of transparency, and also seems to go against the authors’ methodology. Here, the authors mention that FT is understandability after interaction. However, in this study, there is no on-road interaction between the participants and the HMIs. This should be clarified.
3) The authors tested 33 participants. This is a small sample, especially for an on-line study. The manuscript would strongly benefit from the inclusion of a power analysis (e.g. performed via GPower).
4) No significant difference was found between the three tested HMIs. The authors conclude that participants had difficulties understanding the HMIs and that this could be tackled by increasing the “intuitiveness” of the designs (lines 387-392). This conclusion is fairly obvious. The Discussion would benefit from a more extensive reflection on how intuitiveness could be increased.
5) Similarly, the authors conclude that “drivers require some levels of training to understand the automation”. This is not a novel finding, and literature linked to this discussion point should be acknowledged. Furthermore, from the current data, it cannot be assessed if and how experience with a specific Level 2 system impacted the results.
Minor remark:
1) In Procedure, the authors should mention how long it took each participant to complete the experiment.
Given these remarks, I believe that this manuscript should not be published in Information. Nonetheless, I do see merit in the authors’ work. Therefore, I strongly recommend them to tackle these comments and submit this paper elsewhere. As mentioned by the authors, this is a preliminary attempt to assess transparency. Conferences such as AutoUI or CHI may be a valid option. In such venues, discussion with peers could strongly improve follow-up versions of this study.
References:
Kraus, J., Scholz, D., Stiegemeier, D., & Baumann, M. (2020). The more you know: trust dynamics and calibration in highly automated driving and the effects of take-overs, system malfunction, and system transparency. Human factors, 62(5), 718-736.
Oliveira, L., Burns, C., Luton, J., Iyer, S., & Birrell, S. (2020). The influence of system transparency on trust: Evaluating interfaces in a highly automated vehicle. Transportation research part F: traffic psychology and behaviour, 72, 280-296.
Author Response
Please see the attached PDF file.

Round 2
Reviewer 1 Report
I am mostly satisfied with the changes that were made. I have just a few remaining small points, and I recommend that the manuscript be published with minor revisions. As far as I am concerned, no further review is needed.
Minor
The manuscript serves three purposes (lines 119-123): (1) introducing a method to evaluate the transparency of HMI designs; (2) verifying the usefulness of the method using existing HMI designs; (3) identifying which information presented by the HMI design is actually used. For me, this formulation remains unclear about whether and how purpose (3) relates to purpose (1). It appears that purpose (3) in a way is an add-on, and this also weakens the structure of the manuscript: the method and its development are presented mostly clearly; the same applies to the verification; but there is detailed discussion of which information is used. The latter is tedious, considering from the perspective of purposes (1) and (2). On the other hand, in the discussion, the relation is much clearer: “The proposed method would be a suitable design tool to help identify critical information for transparent HMIs”. I would advise the authors to include this perspective, making the link between purposes (1) and (3) explicit, in the presentation of the goals in lines 119-123 (and maybe in the argumentation before lines 119-123 as well). Furthermore, if the authors adopt this advice, I recommend that they look critically at the argumentation of the goals, and build a coherent and compelling argumentation, not limiting themselves to making minimal changes to the manuscript that serve only to superficially address the reviewers' comments – I could not suppress that feeling when going through the revisions made to the original manuscript.
Concerning the questions for the transparency assessment test, which I commented upon in my review for the initial version: the question was how these questions were generated. The authors comment upon this in their response to the reviewers, but the description in the manuscript has not been revised. But if the authors want other researchers to apply this method too, they should help them by providing directions about what questions to ask in the transparency assessment test. In other words, some of the explanation in the authors’ response should go into the manuscript.
Language/grammar
L83: two transparency information: missing text? two transparency information ….?
L227: the availability of sub-systems are used > the availability of sub-systems is used
L264: misleading icons: I would rather say “irrelevant icons” (i.e., icons that are irrelevant to the task at hand”)
L304, formula (2): TF: shouldn’t this be FT?
L334: experiences levels > experience levels
L369ff: The subscript ‘holm’ is unknown or unfamiliar. Please avoid the subscript; instead, explain in the text what you did.
Reviewer 3 Report
I thank the authors for addressing my comments. I believe that the manuscript has strongly improved and I recommend its publication.
